# Retrospective review of the management of acute infections and the indications for antibiotic prescription in primary care in northern Thailand

Rachel C Greer,[1,2] Daranee Intralawan,[3] Mavuto Mukaka,[1,2] Prapass Wannapinij,[1] Nicholas P J Day,[1,2] Supalert Nedsuwan,[3] Yoel Lubell[1,2]

¹Mahidol-Oxford Tropical Medicine Research Unit, Faculty of Tropical Medicine, Mahidol University, Bangkok, Thailand
²Centre for Tropical Medicine and Global Health, Nuffield Department of Medicine, University of Oxford, Oxford, UK
³Social and Preventive Medicine Department, Chiang Rai Regional Hospital, Chiang Rai, Thailand

**Correspondence to**
Rachel C Greer;
rachel@tropmedres.ac

## ABSTRACT

**Introduction** Antibiotic use in low-income and middle-income countries continues to rise despite the knowledge that antibiotic overuse can lead to antimicrobial resistance. There is a paucity of detailed data on the use of antibiotics in primary care in low-resource settings.

**Objective** To describe the presentation of acute infections and the indications for antibiotic prescription.

**Design** A 2-year retrospective review of routinely collected data.

**Setting** All 32 primary care units in one district in northern Thailand.

**Participants** Patients attending primary care with a history of fever, documented temperature, International Statistical Classification of Diseases 10 code for infection or prescribed a systemic antibiotic. Patients attending after the initiation of a study on C-reactive protein testing in four centres were excluded.

**Outcome measures** The proportion of patients prescribed an antibiotic and the frequency of clinical presentations.

**Results** 762 868 patients attended the health centres, of whom 103 196 met the inclusion criteria, 5966 were excluded resulting in 97 230 attendances consisting of 83 661 illness episodes. 46.9% (39 242) of the patients were prescribed an antibiotic during their illness. Indications for antibiotic prescription in the multivariable logistic regression analysis included male sex (adjusted OR (aOR) 1.21 (95% CI 1.16 to 1.28), p<0.001), adults (aOR 1.77 (95% CI 1.57 to 2), p<0.001) and a temperature >37.5°C (aOR 1.24 (95% CI 1.03 to 1.48), p=0.020). 77.9% of the presentations were for respiratory-related problems, of which 98.6% were upper respiratory tract infections. The leading infection diagnoses were common cold (50%), acute pharyngitis (18.9%) and acute tonsillitis (5%) which were prescribed antibiotics in 10.5%, 88.7% and 87.1% of cases, respectively. Amoxicillin was the most commonly prescribed antibiotic.

**Conclusions** Nearly half of the patients received an antibiotic, the majority of whom had a respiratory infection. The results can be used to plan interventions to improve the rational use of antibiotics. Further studies in private facilities, pharmacies and dental clinics are required.

### Strengths and limitations of this study

► Over 80 000 illness episodes reviewed from all primary care units in a district, over a 2-year time period.
► Wide range of infections included rather than focusing on one specific infection.
► Use of routine electronic data (no Hawthorne effect), making this work reproducible.
► Only included public healthcare facilities.
► Reliant on the correct coding and clinical diagnoses of illnesses.

## BACKGROUND

The proportion of global deaths attributable to communicable diseases has greatly reduced in recent years. Despite these improvements, 10.6% of deaths worldwide in 2015 were thought to be caused by lower respiratory tract infections (LRTIs), diarrhoea and tuberculosis (TB).[1] In children under 5 years of age, 51.8% of deaths worldwide were due to infectious causes in 2013, with pneumonia causing 14.8% of the deaths overall.[2] In Thailand in 2010, respiratory infections were the leading cause of hospitalisations and deaths in children under 5 years of age.[3] Prompt access to appropriate antibiotics is vital to prevent many of these unnecessary deaths[4]; but while inappropriate or no treatment remains a clear cause for concern, the global antibiotic consumption rate increased by 39% between 2000 and 2015, fuelled by low-income and middle-income countries (LMICs),[5 6] with the majority of antibiotics being consumed in the community.[7]

Overuse and misuse of antibiotics have been linked to the development of antimicrobial resistance (AMR).[7–9] Antibiotics prescribed to individuals in primary care have been associated with bacterial resistance in that individual for up to 12 months, and longer and

more frequent antibiotic courses are more likely to cause resistance.[10] The WHO has described AMR in Southeast Asia as a 'burgeoning and often neglected' issue, stating that a 'post-antibiotic era' may become reality, resulting in common infections and minor injuries being untreatable.[11] In Thailand in 2010, there were an estimated 19122 deaths attributable to multidrug resistant hospital-acquired infections.[12] Thailand has been making sustained efforts to reduce inappropriate antibiotic use; its Antibiotic Smart Use programme started in 2007 and targets three conditions which are unlikely to require antibiotic treatment but for which they are commonly prescribed: upper respiratory tract infections (URTIs), acute diarrhoea and simple wounds.[13] Prescribing targets have been incorporated into the public health system's pay for performance criteria. In August 2016, the Thai government endorsed a national strategic plan for AMR which aims to optimise antimicrobial drug use and reduce the mean consumption of antimicrobials in humans by 20% by 2021.[14]

To appreciate the scale of the problem and to target future interventions, a greater understanding of the acute infections presenting to primary care and the conditions for which antibiotics are used in LMICs is required. Such data, however, are limited,[15] with most studies deriving their estimates from small samples of health providers and over a limited timeframe, therefore neglecting possible seasonal and spatial variation and other secular trends. In this paper, we describe the proportion of patients receiving an antibiotic prescription and indications for antibiotic use in 32 primary care units (PCUs) across a district in northern Thailand over a 2-year period.

## METHOD
A retrospective computerised search of routinely collected data from PCUs in Mueang Chiang Rai District between January 2015 and December 2016 was carried out.

### Study setting
Thailand is an upper-middle income country. In 2016, its gross domestic product was US$407 billion. The average life expectancy at birth is 75 years.[16] Chiang Rai is the most northern province in Thailand and shares borders with Laos and Myanmar. It has a population of 1 282 544, of whom 241 436 reside in Mueang Chiang Rai District.[17 18] Thailand has three seasons; the wet season typically runs from July to October, the cool season from November to February and the hot season from March to June.

Thailand's healthcare system is made up of public and private providers. Universal health coverage was established in 2002 following significant investment in the healthcare system and infrastructure since the 1970s. In rural and poorer areas, primary healthcare is predominantly provided by the public healthcare system whereas in urban areas hospitals and private clinics play a larger role.[19]

Antibiotics can be bought directly from pharmacies and local stores, as well as being prescribed by healthcare workers. Community antibiotic guidelines exist for some but not all common infections, including assessment criteria (eg, the Centor criteria for acute tonsillitis), first-line antibiotics, their dose and duration. There are prescribing restrictions in place for some broad-spectrum antibiotics such as amoxicillin and clavulanic acid (Co-amoxiclav) which cannot be prescribed by nurses working in the public primary care system. More comprehensive hospital-based guidelines are available.

In Mueang Chiang Rai District, family medicine doctors based at the provincial hospital oversee 32 public PCUs which are staffed primarily by two to five nurses and public health officers. On average, PCUs look after 5000 patients each.[19] They provide care for acute and chronic conditions, as well as providing preventative services such as immunisations, cervical screening and health education. Dental and traditional medicine services are also available. The farthest PCU is 2 hours' drive through the mountains from the provincial hospital in Chiang Rai city (see figure 1). Finger-prick blood glucose tests are the only investigations routinely available on site.

### Inclusion criteria
Patients were identified with at least one of the following:
► Systemic antibiotic prescription.
► International Statistical Classification of Diseases (ICD) 10 code for infection (see online supplementary material, table S1).
► Fever as the chief complaint.
► Documented temperature >37.5°C at the PCU.

We excluded patients attending PCUs used as study sites during or after a recent trial on the use of C-reactive protein (CRP) point of care tests (https://www.clinicaltrials.gov/ct2/show/NCT02758821?term=NCT02758821&rank=1).

### Study outcomes
The primary outcome was the overall proportion of illness episodes prescribed an antibiotic. Risk factors for antibiotic use are reported, and the percentages of patients receiving antibiotics according to their diagnosis, percentages of individual antibiotics used and the frequency and type of acute infection presentations.

### Data collection
With the approval of the Chiang Rai Provincial and Public Health Office (PHO), a research data manager accessed the PHO's routine medical records database to search for relevant patients and extract the prespecified variables. Data collected consisted of the PCU attended, patients' number, age, sex, date of visit, chief complaint, temperature, ICD 10 code and drug prescriptions.

### Data cleaning and coding
The inclusion criteria were classified as being present, absent or that the data were missing. Antibiotics were searched for in the prescription field (free text) and

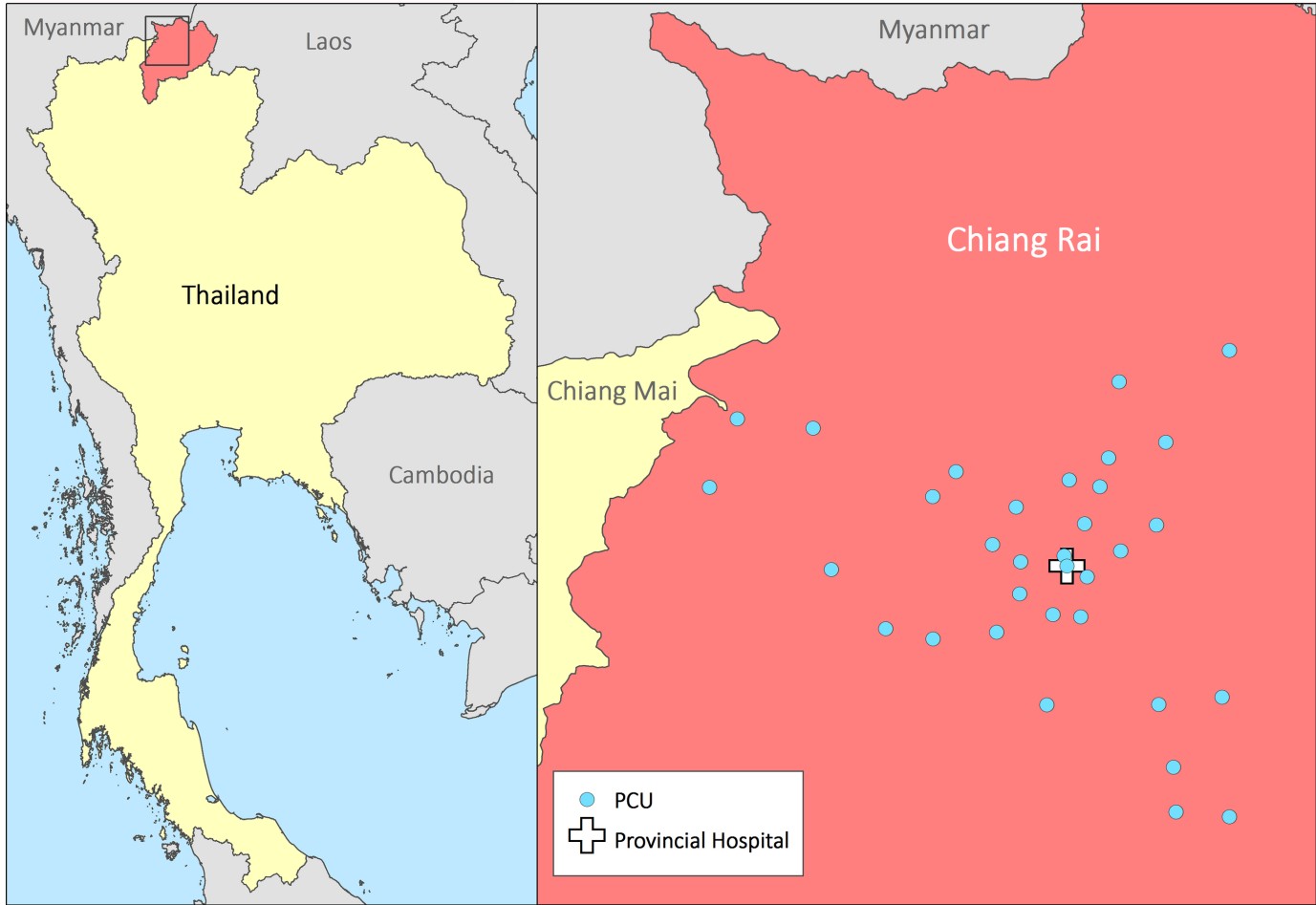

**Figure 1** Chiang Rai and the 32 primary care units (PCUs).

coded as prescribed (yes or no). A predefined antibiotic list (see online supplementary material) was generated using the formulary. All medications prescribed in the dataset were reviewed to ensure no antibiotics were omitted due to spelling errors or their absence from our original list. If no prescription was recorded, we made the assumption that this was because no medication was given rather than the data being missing.

Our predefined list of ICD 10 codes for infection (see online supplementary material, table S1) were searched for in the diagnosis field (free text field, containing ICD 10 codes only) and then coded as present or absent. Our list of ICD 10 codes were reviewed with the other variables to ensure their appropriateness.

We searched for the word 'fever' in Thai in the chief complaint field (free text). In some cases 'no fever' was recorded, or the word 'fever' was present but part of a phrase alluding to a patient more generally, or 'influenza vaccine'. This field was checked manually with the help of two native northern Thai speaking study nurses. History of fever in the chief complaint was then coded as yes or no. Documented temperatures over 37.5°C at the PCU were searched for in the temperature field and then coded as temperature >37.5°C yes or no.

Repeat attendances within 1 month were classed as one illness episode allowing for the detection of subsequent antibiotics or treatment changes. All other indications such as the chief complaint and temperature were taken from the initial presentation. Children were defined as being under 12 years of age. The ICD 10 codes were grouped into gastrointestinal, respiratory, skin, urogenital, eye, ear and other categories. Each category was further broken down into conditions such as acute sinusitis and acute pharyngitis. The respiratory category was also grouped into URTI and LRTI.

## Statistical analyses
### Descriptive statistics

Categorical variables were summarised using counts and percentages. Non-normally distributed data were described using medians and IQRs and compared using the rank-sum test. The proportions of patients prescribed an antibiotic in different demographic groups were summarised and compared using the $\chi^2$ test.

### Logistic and Poisson regression models

A logistic regression model was used to model the binary outcome of antibiotic prescription (yes or no); both

unadjusted and adjusted models were fitted and accounted for clustering of patients attending the same PCU. The ORs for the indications of antibiotic prescription were first obtained from univariate logistic regression models and then considered in a multivariable model if they had a p value of <0.05. In addition to these variables we also included documented temperature because this was considered to be a natural confounder of antibiotic prescription. Indications included sex, age category and documented temperature. The main purpose of this model was to identify risk factors that were independently associated with antibiotic prescription. A temperature of >37.5°C was used rather than the more subjective history of fever. ICD 10 codes were not included because of their strong association with antibiotic prescriptions (eg, a health worker's diagnosis of acute pharyngitis and its affiliated ICD 10 code were inherently associated with antibiotic prescription, as opposed to a diagnosis of 'common cold'). Furthermore, a Poisson regression model of the monthly number of antibiotic prescriptions over a 24-month period was produced to obtain the incidence rate ratios (IRRs) and 95% CIs.

### Time-series analysis

Monthly antibiotic prescriptions were weighted by the number of contributing PCUs per month and modelled over a 2-year period. When time-series analysis is used for forecasting, it is common to apply it to periods of 5 years or more; however, our aim was not to forecast into the future but to simply describe the current trends in antibiotic prescription.[20 21] We used a time-series analysis to separate long-term trends from seasonal variations.[22 23] Symmetric locally weighted moving averages were used. In this procedure, less weight was applied to time points (in months) farthest away from the present time point. The data were available on a monthly basis; however, a quarterly window was used to identify seasonality as follows: $\hat{X}_t = \frac{1}{9}\left(X_{t-2} + 2X_{t-1} + 3X_t + 2X_{t+1} + X_{t+2}\right)$

Similarly, a 12-month time-series window was used to obtain a trend line that would be sensitive to monthly changes but with reduced noise from seasonal variation:

$$\hat{X}_t = \frac{1}{24}\left(X_{t-6} + X_{t+6}\right) + \frac{1}{12}\left(X_t + X_{t-1} + X_{t+1} + X_{t-2} + X_{t+2} + X_{t-3} + X_{t+3} + X_{t-4} + X_{t+4} + X_{t-5} + X_{t+5}\right)$$

Where $X_t$ is the time-series modelled monthly prevalence of antibiotic prescription. Statistical significance was declared at alpha=0.05. Data analyses were performed using STATA V.14.

### Patient and public involvement

Patients were not involved in the design of the study. Due to the study's retrospective nature, patients were not involved in the recruitment processes. Study results will be disseminated through community presentations and educational updates for the healthcare workers and community volunteers.

## RESULTS

A total of 762 868 patients attended the PCUs between 1 January 2015 and 31 December 2016. The majority of patients' attendances included a chronic disease review or screening, the most common being screening for diseases such as diabetes, hypertension, mental health and dental disorders (145 410), essential hypertension reviews (98 822) and routine child health examinations (75 701).

The appropriateness of the ICD 10 codes for infection used in our inclusion criteria were reviewed alongside the other variables. For example, we found that patients with TB, HIV and hepatitis B were only attending for regular medications rather than for acute illnesses so they were removed from the ICD 10 inclusion list. Mass head lice treatment at schools is carried out by the PCUs so these codes were also removed. The ICD 10 code 'K05' (dental) was also removed because it transpired that these patients were seen by dentists or dental nurses at the PCUs rather than by the regular PCU staff. All ICD 10 codes for myositis were removed from the inclusion criteria apart from M60.0 (infective myositis) because the other codes were being used for muscle pain or myalgia (see online supplementary material, table S1).

In total, 103 196 attendances met our inclusion criteria; 5966 were then excluded because the PCUs they attended were involved in the CRP study before or during their attendance, resulting in 97 230 attendances (12.7%) meeting our inclusion and exclusion criteria. A total of 13 569 repeat attendances within 1 month were classed as a single illness episode, leaving 83 661 illness episodes.

### Patient characteristics

The median age was 24 years with an IQR of 6–51 years, two patients had no age recorded and 54.7% of the patients were female (45 779) compared with 45.3% males (37 882) (p<0.001).

The proportion of patients meeting each inclusion criterion is shown in figure 2 and online supplementary material, table S2. A total of 29 246 (35.3%) patients presented with a history of fever, while 10 508 (13.7%) had a temperature of more than 37.5°C at presentation. A total of 8871 (11.6%) patients had both a history of fever and a temperature at presentation.

### Antibiotics

Medications were prescribed for 81 691 (97.7%) illness episodes; 37 011 (44.2%) patients were prescribed an antibiotic during their first visit, and this increased to 39 242 (46.9%) throughout their illness episodes.

Antibiotics were prescribed to:
▶ 49.2% of males compared with 45% of females (p<0.001).
▶ 39% of children compared with 51.8% of adults (p<0.001).
▶ 40.1% of those with a history of fever.
▶ 47.6% with a temperature >37.5°C.
▶ 38.8% with an ICD 10 code for infection.

The proportion of patients within each age group prescribed an antibiotic varied, with the lowest rates in young children (0–4 years old, 33.8%), peaking in adults

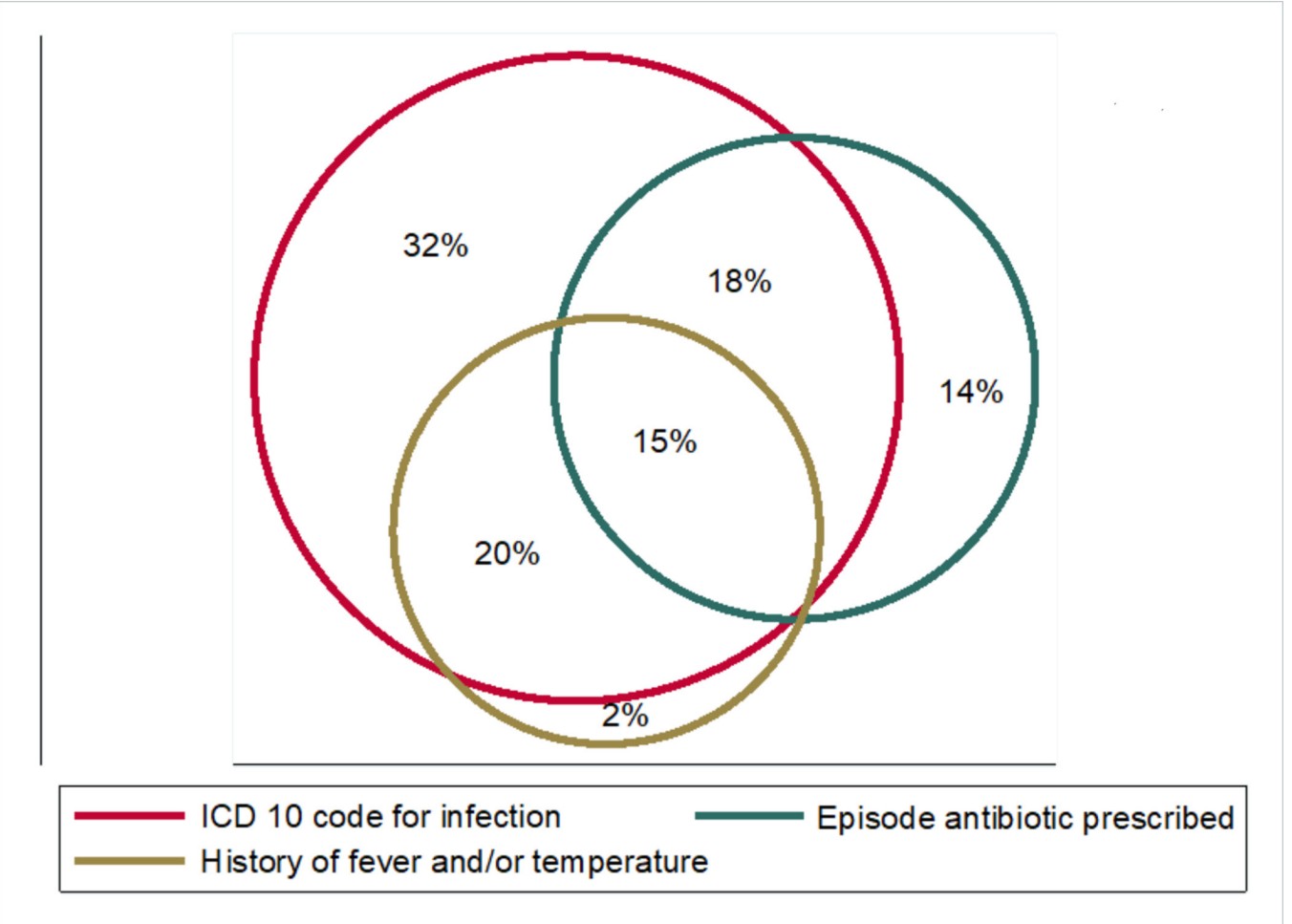

**Figure 2** A Venn diagram to show the inclusion criteria. ICD, International Statistical Classification of Diseases.

(12–39 years old, 55.9%) and then diminishing in the elderly (aged 65 years and older, 41%; see online supplementary material, table S3).

The ORs for the univariate and multivariable logistic regression analyses are shown in table 1. In the univariate model male sex and the adult age category were significant so were added to the multivariable analysis in addition to documented temperature. Indications for antibiotic prescription in the adjusted multivariable logistic regression analysis were male sex (adjusted OR (aOR) 1.21 (95% CI 1.16 to 1.28), p<0.001), patients aged 12 years of age or older (compared with those less than 12 years of age) (aOR 1.77 (95% CI 1.57 to 2), p<0.001)

and having a temperature of more than 37.5°C (aOR 1.24 (95% CI 1.03 to 1.48), p=0.02).

Figure 3 is a time series plot for the monthly prevalence of antibiotic prescriptions. Overall, there was no significant trend; IRR=0.99, 95% CI (0.990 to 1.007), p=0.796, although there is a suggestion of a downward trend beginning in the final 6 months. The monthly prevalence of antibiotic prescriptions was at least 39% throughout the 2-year period. Patients attending in the wet season (July–October) were more likely to receive antibiotics (47.4%) than those attending in the hot and cold seasons (46.6%), p value=0.029. Overall prescription rates varied greatly between the PCUs from 8% to

**Table 1** Univariate and multivariable logistic regression analyses accounting for clustering of patients attending the same PCU, showing all included variables and their association with antibiotic prescription

| Variable | OR (95% CI) | P values | aOR (95% CI) | P values |
|---|---|---|---|---|
| **Univariate analysis** | | | **Multivariable analysis** | |
| Male sex | 1.18 (1.12 to 1.25) | <0.001 | 1.21 (1.16 to 1.28) | <0.001 |
| Aged ≥12 years | 1.68 (1.48 to 1.90) | <0.001 | 1.77 (1.57 to 2) | <0.001 |
| Temperature >37.5°C | 1.05 (0.85 to 1.30) | 0.627 | 1.24 (1.03 to 1.48) | 0.020 |

aOR, adjusted OR; PCU, primary care unit.

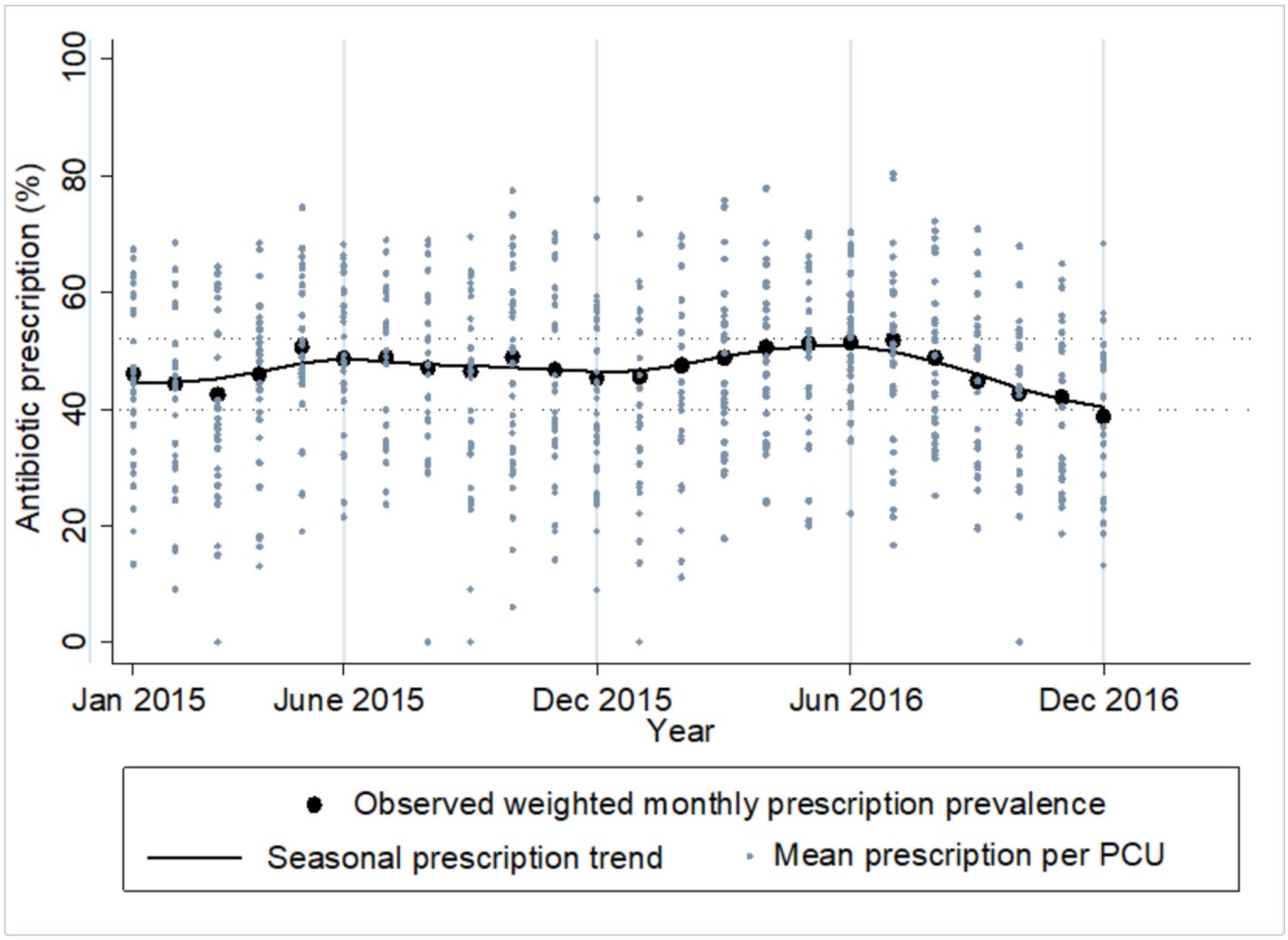

**Figure 3** Trend and seasonality of antibiotic prescriptions overlaid by mean antibiotic prescription rates per primary care unit (PCU).

71.6%, with prescribing consistently higher in adults than in children.

The majority of patients prescribed an antibiotic received amoxicillin (56.7%) or dicloxacillin (25.1%). Other antibiotics prescribed include norfloxacin (8.9%), co-trimoxazole (4.2%), penicillin V (1.2%), roxithromycin (1.2%), metronidazole (1.2%), erythromycin (0.7%), cephalexin (0.4%) and tetracycline (0.2%).

### Presentations and antibiotic prescriptions

The number of acute presentations with ICD 10 codes for infection related to a single system are shown in figure 4; 77.9% of these presentations were for respiratory related problems, 98.6% of these were diagnosed with an URTI, 1.1% with an acute LRTI and 0.3% with a chronic LRTI, of these 36.1%, 81.8% and 53.5% were prescribed antibiotics, respectively. The most common single-infection diagnoses were common cold (34 549, 50%), acute pharyngitis (13 080, 18.9%) and acute tonsillitis (3 459, 5%), antibiotics were prescribed to 10.5%, 88.7% and 87.1% of the cases, respectively (see table 2).

Online supplementary table S4 shows the number of individual infection diagnoses by systems and the rates of antibiotic prescriptions. Antibiotics were prescribed to 59.4% of skin infections, 81.1% of otitis media, 79.5% of otitis externa, 94.7% of cystitis, 80.3% of hordeolum (styes) and chalazions, and 15.7% of conjunctivitis cases. Of the total antibiotics prescribed, almost a third (29.6%) were given to those with acute pharyngitis, followed by common cold (9.3%), acute tonsillitis (7.7%), gastroenteritis and colitis (4.1%), and cystitis (3%) as the single infection diagnoses.

13.8% of patients (11 547) were prescribed antibiotics without a temperature, history of fever or ICD 10 code for infection, of those who had a single diagnosis recorded 1815 (24.6%) of these antibiotics were for dental reasons, 1002 (13.6%) for surgical follow-up care, 526 (7.1%) for contact dermatitis and 473 (6.4%) for open wounds (see online supplementary material, figure S1). These patients were more likely to be male (54.3%, p value <0.001) and older (median age of 41 compared with 24 years) than the main patient group.

### DISCUSSION

To the best of our knowledge, this is the largest review of acute illness presentations and community antibiotic

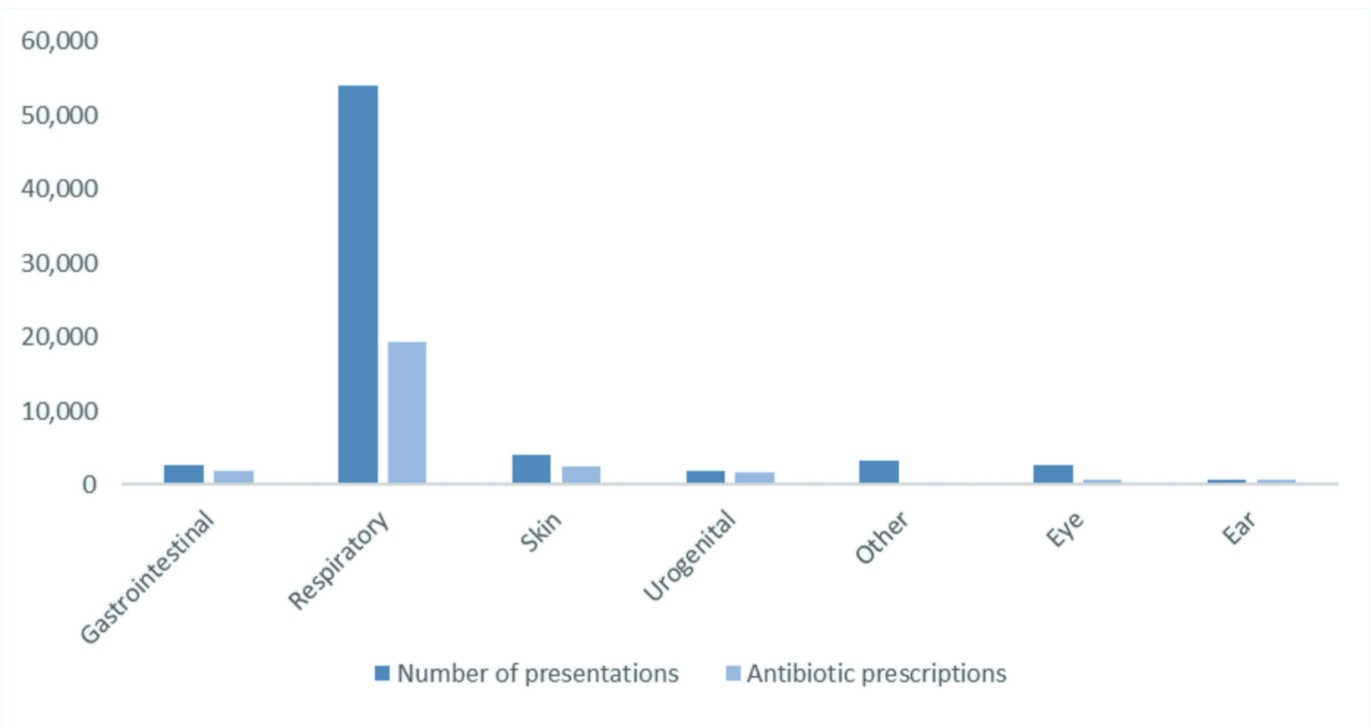

**Figure 4** Number of acute presentations by single systems and whether antibiotics were prescribed.

prescribing in a LMIC. Over a 2-year period, there were more than 97 000 attendances to 32 PCUs for acute infections and nearly half these patients received an antibiotic, with no significant change in prescribing levels over the 2-year study period. Studies of this magnitude are required to increase our knowledge of the scale of antibiotic prescribing and the common conditions they are used for.[24 25] Thailand's 2016 national strategic plan on AMR also highlighted the importance of monitoring and reporting antimicrobial consumption.[26]

Comparing overall antibiotic prescribing rates with other studies is challenging because of varying definitions of acute illnesses and the different patient populations. However, the antibiotic prescribing rate in our study is more than double the prescribing in a Malaysian study but similar to studies in India and Laos.[27–29] A third of our patients had a history of fever which is similar to a point prevalence study in India where fever was the most common symptom.[30] Almost 80% of the ICD 10 codes for infection were related to respiratory infections which is consistent with respiratory infections being the leading cause of hospitalisations and deaths in Thai children under 5 years of age[3] but is higher than the proportion of respiratory presentations in other South and Southeast Asian countries.[30 31] Antibiotic prescribing in Thailand for tonsillitis and pharyngitis remains high despite group A beta-haemolytic *Streptococci* being isolated in only 3.8%–7.9% of those with URTI.[32 33]

In the first phase of Thailand's Antibiotic Smart Use programme, overall antibiotic use in PCUs was reduced by between 39% and 46%. Prescriptions for the three target conditions (URTI, acute diarrhoea and simple wounds) reduced from 54.5% to 25.4%.[34] Despite the lower prescribing levels of 10.5% for common colds in

**Table 2** Common diagnoses in patients with one single International Statistical Classification of Diseases 10 code for infection, whether antibiotics were prescribed and which antibiotic was most commonly used

| Diagnosis | Number of presentations, n/N (%) | Episode antibiotics prescribed, n/N (%) | The most common antibiotic prescribed, name (%) |
|---|---|---|---|
| Common cold | 34 549/69 115 (50) | 3643/34 549 (10.5) | Amoxicillin (71.7) |
| Acute pharyngitis | 13 080/69 115 (18.9) | 11 607/13 080 (88.7) | Amoxicillin (91.5) |
| Acute tonsillitis | 3459/69 115 (5) | 3014/3459 (87.1) | Amoxicillin (93.4) |
| Gastroenteritis and colitis unspecified | 2412/69115 (3.5) | 1614/2412 (66.9) | Norfloxacin (68.8) |
| Conjunctivitis | 2097/69 115 (3.0) | 330/2097 (15.7) | Amoxicillin (56.4) |
| Other helminthiases | 1231/69 115 (1.8) | 65/1231 (5.3) | Amoxicillin (41.5) |
| Cystitis | 1230/69 115 (1.8) | 1165/1230 (94.7) | Norfloxacin (75.9) |

our review, there were still 3643 antibiotic prescriptions for this condition, alongside 88.7% of those with acute pharyngitis, 87.1% with acute tonsillitis and 66.9% with gastroenteritis and colitis receiving antibiotics; this is likely to represent the overuse of antibiotics. Open wounds and superficial injuries were common diagnoses in those prescribed an antibiotic without a history of fever, temperature or ICD 10 code for infection. The results reveal the ongoing high levels of prescribing for these conditions and present an opportunity to further reduce antibiotic use. Since late 2016, an antibiotic prescribing target of less than 20% for these three conditions has been incorporated into Thailand's rational drug use service plan, as well as the pay for performance health criteria, and financial incentives are given to the PCUs achieving this target. A review of the long-term effectiveness of this policy including any impact on patient safety is required.

Our study also identifies high levels of prescribing for skin infections, otitis media, otitis externa, cystitis, hordeolum (styes) and chalazions. A lack of available topical antibiotics may account for the high prescribing for skin infections and otitis externa. However, despite antibacterial eyedrops being available, 15.7% of conjunctivitis cases were still prescribed a systemic antibiotic. Urine dipstick tests are not available on site to assess patients with cystitis or suspected urinary tract infections. Introduction of these simple tests may help to rationalise prescribing for these conditions in a setting where urine cultures are not readily available or achievable.

While we did not set out to review dental prescribing, this area accounted for 25% of the antibiotics prescribed to those without a history of fever, temperature or ICD 10 code for infection which warrants further investigation.

Some of the variation in antibiotic prescribing rates between PCUs may be accounted for by the degree of staff training. Two out of the three highest prescribing PCUs are staffed only by public health officers. The study findings are being used to guide educational updates and training for the PCU staff, with priority being given to those PCUs without nurses and with high prescription rates for conditions unlikely to require antibiotics.

A wide range of antibiotics are prescribed in the PCUs. Restrictions are in place for some broad-spectrum antibiotics such as amoxicillin and clavulanic acid (Co-amoxiclav) which cannot be prescribed. One area of concern is that less than 1% of the antibiotics being prescribed have activity against scrub typhus which is the leading cause of hospital admission with acute undifferentiated fever in this region.[35]

### Strengths and limitations

The main strength of this study is the large number of illness episodes included. The 2-year time period should allow for seasonal variations and disease epidemics. We reviewed prescribing in all of the PCUs in Mueang Chiang Rai District which covers a large geographical area and has a range of rural and urban facilities, making the results generalisable to the region more broadly. Many studies have focused on prescribing for specific conditions such as URTIs but our study covers a wide range of infections that present in the community. Having research staff on site has been shown to influence healthcare workers' prescribing habits (the Hawthorne effect), but due to the retrospective nature of the study this was not a source of bias. The use of routinely collected data means that this methodology could be repeated in other districts and provinces in Thailand, although a lot of the data are entered as free text which presents challenges for analysis. Searching for patients with a history of fever, for instance, was problematic because the Thai word 'ไข้' or fever is also part of the Thai words for patient, influenza, antipyrexials and so on.

Limitations of this study are that we only included public PCUs and have no data on antibiotic use by private clinics, pharmacies or family medicine doctors based at the provincial hospital which requires further study. The PCU data are taken from routine electronic records and in some instances there were tranches of missing data (five PCUs had no recorded data for several months). Verifying the quality of some data is also challenging; coding of clinical diagnoses for instance using ICD 10 could be inconsistent between healthcare workers and in primary care the majority of infections are diagnosed clinically without any laboratory tests. However, we used data from a subsample of patients enrolled in a clinical trial in four PCUs and compared them with their respective routine medical records. While minor discrepancies were found in their precise age and temperature, the diagnoses and antibiotic prescribing data were consistent. Our decision to class all attendances within a 1-month period as a single illness episode means that we may have incorrectly classed some new illnesses as a repeat attendance but did allow us to review antibiotic prescribing over the course of the illness. The time series analysis was carried out using data from a 2-year time period, more definitive conclusions and trends may have become apparent if further time points and data were available.

## CONCLUSIONS

This study provides much needed insight into the use of antibiotics in primary care in northern Thailand, allowing targeting of interventions to improve the rational use of antibiotics. Nearly half of all patients attending with an acute illness received an antibiotic. The majority of presentations were for respiratory infections. Further education and resources are required to support clinicians in the targeting of antibiotics. This could include the introduction of clinical algorithms and point-of-care tests such as CRP and urine dipsticks. Antibiotic guidelines are required for common conditions seen in primary care outside of the current Antibiotic Smart Use policy. Further studies, including qualitative work, are required to appreciate the use of antibiotics in other settings such as private facilities, pharmacies and dental clinics.

**Acknowledgements** The authors would like to thank Wasuphal Kanjana for his help with data collection, Nipaphan Kanthawang and Pratakpong Wongkiti for their help with data cleaning and Areerat Thaiprakhong for her assistance with figure 1.

**Contributors** All authors were involved in the design of the study. PW collected the data. RCG carried out the analysis with support from YL. MM provided statistical support. RCG, YL, DI and SN interpreted the data. RCG wrote the first draft of the paper. YL, NPJD and MM reviewed subsequent drafts. All authors approved the final draft for publication.

**Funding** This study was part of the Wellcome Trust Major Overseas Programme in SE Asia (grant number 106698/Z/14/Z).

**Competing interests** None declared.

**Patient consent** Not required.

**Ethics approval** Ethical approval was obtained from Chiang Rai's Provincial and Public Health Office IRB (number 56/2560). Exemption was given by the Oxford Tropical Research Ethics Committee (OxTREC).

**Provenance and peer review** Not commissioned; externally peer reviewed.

**Data sharing statement** We are unable to share additional unpublished data which falls under the jurisdiction of the Chiang Rai PHO.

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
