## [Reviewer comments · BMJ Open]

ARTICLE DETAILS

TITLE (PROVISIONAL)	A Retrospective Review of the Management of Acute Infections and the Indications for Antibiotic Prescription in Primary Care in Northern Thailand
AUTHORS	Greer, Rachel; Intralawan, Daranee; Mukaka, Mavuto; Wannapiniij, Prapass; Day, Nicholas; Nedsuwan, Supalert; Lubell, Yoel

VERSION 1 – REVIEW

REVIEWER	Mamoon Aldeyab- Lecturer in Clinical Pharmacy Ulster University -UK
REVIEW RETURNED	11-Mar-2018

GENERAL COMMENTS	The authors attempted to describe the presentation of acute infections and the indicators for antibiotic prescriptions, over a two year retrospective survey, in Northern Thailand. The work is of value, relevance and importance. To improve the clarity of the manuscript, I have the following comments: - On page 4, lines 46 and 47: In this section, the authors stated” if no prescription was recorded we maderather than data being missing”. However, on page 9 lines 30 to 31, the authors stated that there were missing data for several months “the PCU data is taken from there were trenches of missing data”. Please clarify- Regarding reported statistics on page 4, it is difficult to follow reported material in this section. Please construct a section, name it “statistical analyses”, and report related statistics addressing the following content: (i) descriptive statistics, (ii) logistic regression, (iii) time-series analysis. • Regarding logistic regression, please state clearly the name of the outcome/response variable and how was this coded (yes/no), and then state the name of the independent variables, the p-value that was considered for the univariate logistic regression, and what was the nature of the conducted regression (e.g. backward analysis). Also, please state the reason for carrying out this analysis.• Regarding time-series analysis: please state clearly the reason why you used time-series analysis, what components were used (e.g. AR, MA, ARIMA models), and a reference for the used time-series analysis methodology- Would two years be enough to do time-series analysis? Was your series measured as monthly, quarterly, etc.? - Under the methods section, can you please indicate what antibiotic guidelines are available in community and briefly describe it- On page 6, line 45: how was the “incidence rate ratio” calculated? Please describe it in the methods section.- On page 6, line 44, the observed no significance trend should be
--

	interpreted in caution as possibly there were no enough data points (usually for time series analysis, 5 years of date (60 observations/monthly data)) is required.  - On page 6, line 46, and regarding “ the observed downward trend observed in the last 6 months”, there was also a similar (however slight) downward trend observed for June 2015 to Dec 2015. More data points is needed to give definitive conclusion. - On page 6, lines 51 to 54, there is no mentioning for (amoxicillin plus clavulanic acid)- is it not used at all? If it's used, what is the percentage of using this antibiotic? - On page 7, within the presented table, antibiotic for treating common cold was 10.5%, and there are other different percentages for certain medical conditions on line 39 to 45- please discuss relevant antibiotic misuse in the discussion section. - On page 7, lines 46-47, i.e. “the distribution of ages Were both significantly different”- what is the relevance/context for this statement? - In the results section, please present Tables for the univariate analysis and multivariate analysis. - On a related note to the first point, and considering the reported material in the limitation section, page 9, lines: 28-36, the obtained data suffered from missing data, inaccuracy or inconsistency in coding of clinical diagnosis using ICD10- how did the authors deal with these points and how was the quality of data verified? was the original dataset, that these data were taken from, validated and verified?
--	--

REVIEWER	Pawel Stefanoff National Institute of Public Health, Poland
REVIEW RETURNED	15-Mar-2018

GENERAL COMMENTS	The manuscript entitled "A Retrospective Survey of the Management of Acute Infections and the Indicators for Antibiotic Prescription in Primary Care in Northern Thailand" is a very valuable contribution to one of the Public Health priority areas. The numbers speak for themselves, and the authors address this by focusing on clear reporting, without too much unnecessary interpretation. I just have some suggestions to improve even further the clarity of the manuscript:  1. I'm not a big fan of labelling all research, but sometimes it can be beneficial for clarity. The authors use the term "survey" suggesting that they have conducted a cross-sectional study. Since there was no sampling involved, maybe it would be sufficient just to label the study "retrospective chart review" (for consideration, the methods are clear, and you could still call it a cross-sectional survey considering the region being a sample of the country population). 2. It is important to mention the Smart Use Policy in the background, as setting the scene for the study. 3. The first two paragraphs of the methods should be part of the background. 4. There is small confusion in the "study outcomes" section. First, this was not a rate (measure of time in the denominator) but a ratio or proportion. Second, I think that the main outcome was rather the administration of antibiotic prescription (a binary outcome, yes/no), and the proportion of particular subgroups that were prescribed antibiotics were indicators, rather than outcomes. Similarly, the odds ratio is an epidemiological indicator. It is crucial to clarify this in the methods and in the abstract. 5. Page 5, lines 9-13: This is not a good place to discuss the
--

	definitions of the variables. You should in fact secure a separate section where you would move the descriptions of the variables definition, categorisation, recoding, including whether they were initially coded at data entry or taken and categorised based on the open text fields. Maybe even a table would be helpful here that would clearly summarise the variable source and categorisation. 6. Page 5, lines 48-56: Consider turning into a flowchart summarising the exclusion of particular syndromes. These are the key observations I had. I found the discussion very focused with such broad and important area - not an easy task! Thanks for this contribution!!!
--	--

VERSION 1 – AUTHOR RESPONSE

Reviewer(s)' Comments to Author:

Reviewer: 1

Reviewer Name: Mamoon Aldeyab

Institution and Country: Lecturer in Clinical Pharmacy, Ulster University -UK

Please state any competing interests or state 'None declared': None

Please leave your comments for the authors below The authors attempted to describe the presentation of acute infections and the indicators for antibiotic prescriptions, over a two year retrospective survey, in Northern Thailand. The work is of value, relevance and importance.

To improve the clarity of the manuscript, I have the following comments:

- On page 4, lines 46 and 47: In this section, the authors stated" if no prescription was recorded we maderather than data being missing". However, on page 9 lines 30 to 31, the authors stated that there were missing data for several months "the PCU data is taken from there were trenches of missing data". Please clarify

The section on page 4 is referring to cases where no prescriptions were recorded (in 2.3% of the records). Page 9 refers to multiple fields or no data at all being present. Page 9, line 31 has been changed to '5 PCUs had no recorded data for several months'.

- Regarding reported statistics on page 4, it is difficult to follow reported material in this section. Please construct a section, name it "statistical analyses", and report related statistics addressing the following content: (i) descriptive statistics, (ii) logistic regression, (iii) time-series analysis.

Headings added and text expanded as appropriate

• Regarding logistic regression, please state clearly the name of the outcome/response variable and how was this coded (yes/no), and then state the name of the independent variables, the p-value that was considered for the univariate logistic regression, and what was the nature of the conducted regression (e.g. backward analysis). Also, please state the reason for carrying out this analysis.

This has now been clarified in the logistic regression section. We have also clarified the descriptive statistics

• Regarding time-series analysis: please state clearly the reason why you used time-series analysis, what components were used (e.g. AR, MA, ARIMA models), and a reference for the used time-series analysis methodology- Would two years be enough to do time-series analysis? Was your series measured as monthly, quarterly, etc.?

We have clarified that we used a symmetric locally weighted MA model. In terms of time, we agree with the reviewer that often periods of 5 years or more are more informative for forecasting in time series analysis than shorter periods. However, the aim of our time-series analysis model was not to forecast into the future but rather to describe the current situation hence we applied it to time that is less than 5 years. There are a few publications that have done a similar short period time series analysis, probably for the same purpose as ours. We have included some references as advised.

We have also clarified that the measurements were done monthly but the model adjusted for quarterly seasons.

- Under the methods section, can you please indicate what antibiotic guidelines are available in community and briefly describe it

Details of the Antibiotic Smart Use program and community antibiotic guidelines have been added to the background section (the study site section has been moved to the introduction in line with reviewer 2's comments.)

- On page 6, line 45: how was the "incidence rate ratio" calculated? Please describe it in the methods section.

We agree, we have now included a description in the text that a Poisson regression model was used to obtain the incidence rate ratios and the 95% confidence intervals. This is in the section called Logistic and Poisson regression models

- On page 6, line 44, the observed no significance trend should be interpreted in caution as possibly there were no enough data points (usually for time series analysis, 5 years of date (60 observations/monthly data)) is required.

We have added a sentence to the limitations section of the discussion to clarify this

- On page 6, line 46, and regarding "the observed downward trend observed in the last 6 months", there was also a similar (however slight) downward trend observed for June 2015 to Dec 2015. More data points is needed to give definitive conclusion.

We have added a sentence to the limitations section of the discussion to clarify this

- On page 6, lines 51 to 54, there is no mentioning for (amoxicillin plus clavulanic acid)- is it not used at all? If it's used, what is the percentage of using this antibiotic?

It is not in the formulary for the PCUs so is not prescribed. (It can be prescribed by doctors based in the hospital clinics). See the discussion section

- On page 7, within the presented table, antibiotic for treating common cold was 10.5%, and there are other different percentages for certain medical conditions on line 39 to 45- please discuss relevant antibiotic misuse in the discussion section.

Discussion expanded:

'Despite the lower prescribing levels of 10.5% for common colds in our review there were still 3,643 antibiotic prescriptions for this condition, alongside 88.7% of those with acute pharyngitis, 87.1% with acute tonsillitis and 66.9% with gastroenteritis and colitis receiving antibiotics, this is likely to represent the overuse of antibiotics.'

- On page 7, lines 46-47, i.e. "the distribution of ages Were both significantly different"- what is the relevance/context for this statement?

We agree with you that this statement adds little to the paper, so it has been removed

- In the results section, please present Tables for the univariate analysis and multivariate analysis.

This table has been added

- On a related note to the first point, and considering the reported material in the limitation section, page 9, lines: 28-36, the obtained data suffered from missing data, inaccuracy or inconsistency in coding of clinical diagnosis using ICD10- how did the authors deal with these points and how was the quality of data verified? was the original dataset, that these data were taken from, validated and verified?

The original electronic database has been validated and verified against data collected onsite by study nurses. We can confirm that there are very few errors in the key variables such as history of fever and antibiotic prescription.

Due to the retrospective nature of this review we were unable to independently verify the clinical diagnoses made but the recording of the coding is accurate. The majority of diagnoses relating to

infections presenting to primary care are made clinically without any laboratory diagnostics, so we feel that this data reflects normal clinical practice.

Reviewer: 2

Reviewer Name: Pawel Stefanoff

Institution and Country: National Institute of Public Health, Poland

Please state any competing interests or state 'None declared': None declared

Please leave your comments for the authors below The manuscript entitled "A Retrospective Survey of the Management of Acute Infections and the Indicators for Antibiotic Prescription in Primary Care in Northern Thailand" is a very valuable contribution to one of the Public Health priority areas. The numbers speak for themselves, and the authors address this by focusing on clear reporting, without too much unnecessary interpretation.

I just have some suggestions to improve even further the clarity of the manuscript:

1. I'm not a big fan of labelling all research, but sometimes it can be beneficial for clarity. The authors use the term "survey" suggesting that they have conducted a cross-sectional study. Since there was no sampling involved, maybe it would be sufficient just to label the study "retrospective chart review" (for consideration, the methods are clear, and you could still call it a cross-sectional survey considering the region being a sample of the country population).

Changed 'survey' to 'review'

2. It is important to mention the Smart Use Policy in the background, as setting the scene for the study.

Agreed and added

3. The first two paragraphs of the methods should be part of the background.

The section 'study sites' has been moved to the background section

4. There is small confusion in the "study outcomes" section. First, this was not a rate (measure of time in the denominator) but a ratio or proportion. Second, I think that the main outcome was rather the administration of antibiotic prescription (a binary outcome, yes/no), and the proportion of particular subgroups that were prescribed antibiotics were indicators, rather than outcomes. Similarly, the odds ratio is an epidemiological indicator. It is crucial to clarify this in the methods and in the abstract.

We agree that this section was rather confusing. The primary outcome has been changed to the proportion of patients prescribed an antibiotic. We agree that the use of 'indicators' was confusing and instead refer to 'indications' or 'risk factors' for antibiotic use meaning factors that influence antibiotic prescription, which are not outcomes in their own right. Hopefully this has provided enough clarity in the manuscript.

5. Page 5, lines 9-13: This is not a good place to discuss the definitions of the variables. You should in fact secure a separate section where you would move the descriptions of the variables definition, categorisation, recoding, including whether they were initially coded at data entry or taken and categorised based on the open text fields. Maybe even a table would be helpful here that would clearly summarise the variable source and categorisation.

We agree that this section was not very clear. The heading 'Data cleaning and coding' has been added with further explanation of the variables and how they were coded. A list of antibiotics has been added to the supplementary material alongside the ICD codes table

6. Page 5, lines 48-56: Consider turning into a flowchart summarising the exclusion of particular syndromes.

Added a reference link to supplementary table S1 to highlight the excluded ICD 10 codes and the numbers removed per code was added. It would be challenging to display this as a flow chart because some patients had more than one diagnosis/ICD 10 code for infection, so even if one ICD 10 code was reclassified as not meeting the inclusion criteria they could still meet it for their second diagnosis or meet another inclusion criteria such as antibiotic prescription.

These are the key observations I had. I found the discussion very focused with such broad and important area - not an easy task! Thanks for this contribution!!!

VERSION 2 – REVIEW

REVIEWER	Mamoon Aldeyab Ulster University, UK
REVIEW RETURNED	25-Apr-2018

GENERAL COMMENTS	Authors addressed appropriately my comments- I have no other comments
---

REVIEWER	Pawel Stefanoff Norwegian Institute of Public Health, Oslo, Norway
REVIEW RETURNED	09-May-2018

GENERAL COMMENTS	The manuscript has been greatly improved. Please find below some minor suggestions: 1. The section describing the "Study sites" has been moved to the background. Although I think the background is a better place to describe the background on Thailand, it shouldn't be labeled "study sites" as it suggests being part of the method. 2. At the end of the background the aim should be stated. It should constitute a bridge between the background information (to somehow justify why you decided to present the specific info) and the methods... 3. Small typo on page 8, line 3: "inclusion criterion" and not "criteria"
---

VERSION 2 – AUTHOR RESPONSE

Thank you for your decision letter and for your comments and those of the reviewers. We're delighted to hear that our manuscript has been recommended for publication.

The sub-section "Study sites" has been renamed "Study setting" and moved to the methods section as recommended by the editor. I hope this sufficiently addresses reviewer 2's first comment.

This means that the final paragraph of the background details the research gap and I have added to the final sentence to try and articulate our aims clearly.

Comment 3: thank you for spotting this, "criteria" has been changed to "criterion" in the patient characteristics section.